# Association between optical coherence tomography-defined culprit morphologies and changes in hyperemic coronary flow after elective stenting assessed by transthoracic Doppler echocardiography

Eisuke Usui[1], Yoshihiro Hanyu[1], Tatsuya Sakamoto[1], Masahiro Hoshino[1], Masahiro Hada[1], Tatsuhiro Nagamine[1], Kai Nogami[1], Hiroki Ueno[1], Mirei Setoguchi[1], Kazuki Matsuda[1], Kodai Sayama[1], Tomohiro Tahara[1], Takashi Mineo[1], Yoshihisa Kanaji[1], Tomoyo Sugiyama[2], Taishi Yonetsu[2], Tetsuo Sasano[2], Tsunekazu Kakuta[1] *

1 Cardiovascular Medicine, Tsuchiura Kyodo General Hospital, Tsuchiura, Ibaraki, Japan, 2 Department of Cardiovascular Medicine, Tokyo Medical and Dental University, Tokyo, Japan

* kaz@joy.email.ne.jp

## Abstract

### Background

Stress-transthoracic Doppler echocardiography (S-TDE) provides a noninvasive assessment of coronary flow parameters in the left anterior descending artery (LAD). However, the association between morphological characteristics and coronary flow changes after elective percutaneous coronary intervention (PCI) remains unclear. We aimed to evaluate the relationships between periprocedural coronary flow changes observed on S-TDE and lesion-specific plaque characteristics obtained by optical coherence tomography (OCT) in the interrogated vessels in patients with chronic coronary syndrome (CCS).

### Methods and results

Patients with CCS who underwent pre- and post-PCI S-TDE and elective fractional flow reserve (FFR)-guided PCI under OCT guidance for *de novo* single LAD lesions were included. S-TDE-derived hyperemic diastolic peak flow velocity (hDPV) was used as a surrogate for coronary flow. Lesions were categorized into two groups based on the %hDPV increase or decrease. The baseline clinical, physiological, and OCT findings were compared between the groups. In total, 103 LAD lesions were studied in 103 patients. After PCI, hDPV significantly increased from 55.6 cm/s to 69.5 cm/s (P<0.01), with a median %hDPV increase of 27.2 (6.32–59.1) %, while %hDPV decreased in 20 (19.4%) patients. The FFR improved in all patients. On OCT, layered plaques were more frequently present in the culprit vessels in the %hDPV-decrease group than in the %hDPV-increase group (85.0% vs. 50.6%, P = 0.01). Multivariable logistic regression analysis showed that the presence of layered plaques and high pre-PCI hDPV were independent predictors of %hDPV decrease.

**Data Availability Statement:** All relevant data are within the paper and its Supporting Information files.

**Funding:** The author(s) received no specific funding for this work.

**Competing interests:** The authors have declared that no competing interests exist.

## Conclusions

In patients who underwent successful uncomplicated elective PCI for *de novo* single LAD lesions, the presence of layered plaques was independently associated with hyperemic coronary flow decrease as assessed by S-TDE.

## Introduction

Fractional flow reserve (FFR) is the standard metric for evaluating myocardial ischemia based on the severity of epicardial coronary artery stenosis. As inducible ischemia has been associated with worse clinical outcomes [1–3] an increase in coronary flow to the ischemic region is the fundamental expectation for revascularization. However, elective percutaneous coronary intervention (PCI) does not necessarily increase coronary flow during maximum hyperemia [4–6] Given that previous large-scale clinical trials have shown no benefit of PCI on long-term mortality in patients with chronic coronary syndrome (CCS) [7, 8] contemporary indications for elective PCI in previous studies might not optimize coronary flow improvement. The benefit of PCI, which does not increase coronary flow, is questionable and may even pose harm to patients with procedural complications, stent-related events, and a high bleeding risk with dual antiplatelet therapy. To date, limited studies have reported on changes in coronary flow after PCI [6, 9–12] thus, predictors of coronary flow changes should be investigated to identify patients who may benefit from myocardial perfusion via elective revascularization.

Stress-transthoracic Doppler echocardiography (S-TDE) is a non-invasive, widely available, and cost-effective modality that provides coronary flow velocity without requiring radiation exposure, contrast medium, or catheterization [6] Optical coherence tomography (OCT) provides high-resolution images and can be used to identify high-risk anatomical plaque features that have been established as fundamental in the pathogenesis of coronary artery disease [13, 14] We hypothesized that changes in hyperemic coronary flow after PCI are associated with functional stenosis severity and culprit lesion morphology. Hence, this study aimed to evaluate the relationships between periprocedural coronary flow changes observed on S-TDE and lesion-specific plaque characteristics obtained by OCT in the interrogated vessels, as well as the FFR, thereby identifying predictors of hyperemic coronary flow changes.

## Materials and methods

### Study population

We performed a retrospective analysis of pooled data from the institutional imaging and physiology registry at Tsuchiura Kyodo General Hospital by identifying patients with CCS who underwent elective FFR-guided PCI (symptomatic and FFR ≤0.80) with OCT examinations before PCI between April 1, 2019 and July 31, 2023. In total, 523 vessels from 452 patients who underwent pre- and post-PCI S-TDE examinations were identified for analysis. Patients with following characteristics were excluded: underwent stenting for the left main disease alone (n = 4), right coronary artery (n = 113), left circumflex artery (n = 100), and bypass graft (n = 2); in-stent lesions (n = 46), insufficient guidewire-based physiological data (n = 35), suboptimal S-TDE-derived physiological data (n = 36), and clinically relevant periprocedural myocardial infarction, as defined by an expert consensus document from the Society for Cardiovascular Angiography and Interventions (n = 8) [15] underwent ballooning or atherectomy prior to OCT (n = 44); underwent remote stenting for tandem lesions (n = 2); insufficient

The institutional OCT database in Tsuchiura Kyodo General Hospital between April 2019 and July 2023

```
┌────────────────────────────────────────────────────────────┐
│ 523 vessels in 452 pts with CCS who underwent elective OCT-guided PCI │
└────────────────────────────────────────────────────────────┘
        Exclusion criteria
        • Right coronary artery (n=113)
        • Left circumflex artery (n=100)
        • Left main disease alone (n=4)
        • Graft (n=2)
┌────────────────────────────────────────────────────────────┐
│ 304 vessel in 304 pts with CCS who underwent elective OCT-guided PCI for LAD lesions │
└────────────────────────────────────────────────────────────┘
        Exclusion criteria
        • In-stent lesions (n=46)
        • Lacking or insufficient wire-based physiology data (n=35)
        • Lacking or insufficient stress-transthoracic Doppler echocardiography-derived physiology data (n=36)
        • Type 4A myocardial infarction (n=8)
        • Ballooning or atherectomy before OCT imaging (n=44)
        • Tandem stenting (n=2)
        • Insufficient OCT imaging (n=3)
        • Atrial fibrillation at the time of stress-transthoracic Doppler echocardiography (S-TDE) (n=4)
        • Multivessel significant stenosis at the time of S-TDE (n=14)
        • Hemodialysis (n=9)
┌────────────────────────────────────────────────────────────┐
│ 103 vessels in 103 pts with CCS who underwent elective FFR- and OCT-guided PCI for single LAD │
│ lesions and pre & post-PCI S-TDE assessments │
└────────────────────────────────────────────────────────────┘
```

| Decrease group | Increase group |
|---|---|
| Hyperemic DPV increase <0% | Hyperemic DPV increase >0% |
| N=20 | N=83 |

**Fig 1. Study population.** CCS, chronic coronary syndrome; FFR, fractional flow reserve; DPV, diastolic peak velocity; LAD, left anterior descending artery; OCT, optical coherence tomography; PCI, percutaneous coronary intervention; S-TDE, stress-transthoracic Doppler echocardiography.

OCT data (n = 3); patients with severe chronic kidney disease (creatine level >2.0 mg/dL) (n = 9); those with atrial fibrillation at the time of S-TDE measurement (n = 4); and those with multivessel disease at the time of PCI for left anterior descending (LAD) artery (n = 14). Therefore, the final dataset included 103 LAD lesions from 103 patients (Fig 1). All the patients received guideline-directed medical therapy before undergoing PCI. Authors had access to information that could identify individual patients during and after data collection. The data were accessed on August 4, 11 and 18, 2023 for research purposes.

The study protocol was approved by our institutional review board (2022FY141). This study complied with the principles of the Declaration of Helsinki for human exploration. Written informed consent from the patients enrolled in this study was waived based on the optout method facilitated through our hospital bulletin board.

## Cardiac catheterization procedure

Each patient initially underwent standard selective coronary angiography via the radial artery, using a 6-F system. Coronary angiograms were quantitatively analyzed using the QAngio XA system (Medis Medical Imaging Systems, Leiden, Netherlands) to measure the minimal and reference lumen diameters, percent diameter stenosis, and target lesion length. All patients received a bolus injection of heparin (5,000 IU) before the procedure, and an additional bolus injection of 2,000 IU was administered every hour as needed. An intracoronary bolus injection of nitroglycerin (0.2 mg) was administered at the start of the procedure and repeated before the imaging and physiological assessments. All patients underwent successful and uncomplicated coronary 2nd-generation drug-eluting stent implantation. To avoid aggressive stent expansion, online quantitative coronary angiography was performed to select the appropriate stent size. Successful PCI was defined as <20% residual stenosis with a grade 3 Thrombolysis In Myocardial Infarction (TIMI) flow.

## Physiological assessment

FFR was determined for vessels that were clinically indicated for evaluation because they presented with angiographic intermediate stenosis, using a RadiAnalyzer Xpress instrument with a Pressure Wire (Abbott Vascular, St. Paul, MN, USA). After nitroglycerin administration, a pressure-monitoring guidewire was advanced distally to the stenosis. Hyperemia was induced by an intravenous infusion of adenosine 5'-triphosphate (160 μg/kg/min). The FFR was calculated by dividing the mean distal pressure ($P_d$) by the mean aortic pressure ($P_a$) during steady-state maximal hyperemia. After FFR assessment, when the pressure sensor reached the tip of the guiding catheter during hyperemia via a pull-back maneuver, a mean $P_d$–$P_a$ pressure drift of $\leq 2$ mmHg was documented. We mandated repeat assessment if the pressure drift was $> 2$ mmHg. All patients were instructed to strictly refrain from caffeinated beverages for $> 24$ hours before catheterization.

## OCT image acquisition and analysis

OCT was performed before any interventional procedure using a frequency-domain OCT system (ILUMIEN OPTIS, Abbott Vascular, Santa Clara, CA, USA). The technique used for OCT has been described previously [13, 14] As a frequency-domain OCT system, a 2.7-Fr (Dragonfly OPTIS or Dragonfly Opstar, Abbott Vascular) catheter was advanced over a guidewire, followed by automated pullback at a speed of 18–36 mm/s and 180 frames/s with continuous contrast injection (4 mL/s, 14–18 mL total). The data were digitally stored and analyzed offline. All OCT images were analyzed using proprietary software at the Tsuchiura Kyodo General Hospital OCT Laboratory. Qualitative parameters were assessed within the segment from 20 mm proximal to 20 mm distal to the minimal lumen area by two experienced investigators (E. U. and M.H.) who were blinded to the patients' clinical demographics and angiographic, physiological, and echocardiographic data. Discordance between the two investigators was resolved by consensus. The lumen contour was semi-automatically traced on cross-sectional images using proprietary software and manually corrected by the investigator, if needed. The cross-sectional area with the smallest lumen area was defined as the minimal lumen area.

The plaques were categorized as lipidic, fibrous, or calcified. A lipidic plaque had a region with strong signal attenuation and diffuse border and was considered lipid-rich if the lipidic angle was >90˚. Plaque rupture was defined as a disrupted fibrous cap with intraplaque cavity formation. Thin-cap fibroatheroma had a fibrous cap thickness <80 μm with lipid-rich plaque. The fibrous plaques had homogeneous signal-rich regions. Calcified plaque was defined as a signal-poor or heterogeneous region with sharply delineated borders. Cholesterol crystal was defined as a thin linear region of high intensity having a clear border, with adjacent tissue that was not present within or at the border of the calcified plaque.[16] A layered plaque was defined as a layer of tissue located close to the luminal surface with clear demarcation from the underlying plaque, which was categorized as lipidic (Fig 2A) or non-lipidic (Fig 2B) [17] Macrophages were defined as signal-rich, distinct, or confluent punctate regions accompanied by heterogenic signal shadows. Microvessels were non-signal tubuloluminal structures in a plaque that was not connected to the lumen [18].

## S-TDE acquisition and analysis

Patients underwent pre- (1 day before) and post-procedural (3 days after) LAD coronary flow assessments using S-TDE. Echocardiographic studies were performed according to the American Society of Echocardiography guidelines [19] using a commercially available digital ultrasound system (GE Vivid E95; GE Vingmed Ultrasound, Horten, Norway) with a multifrequency transducer and second-harmonic technology. After the standard examination,

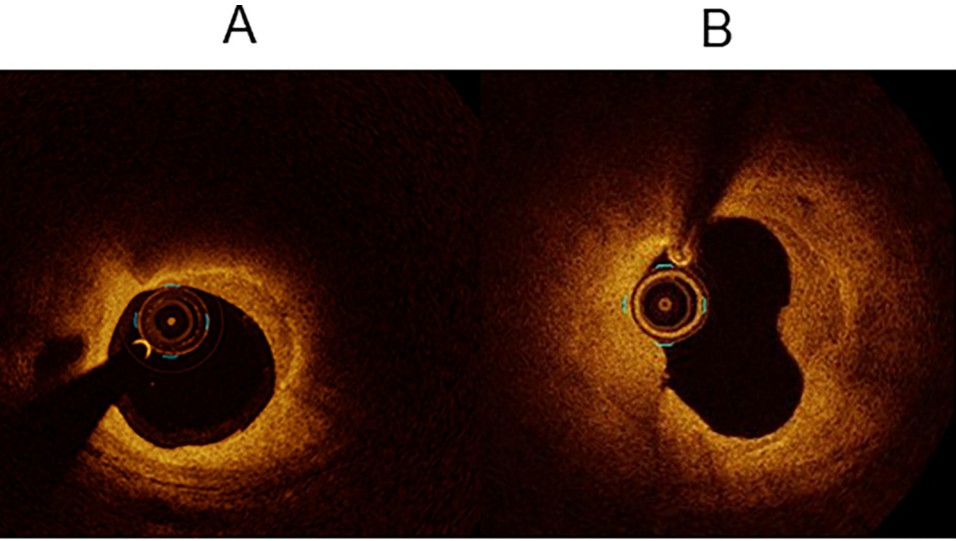

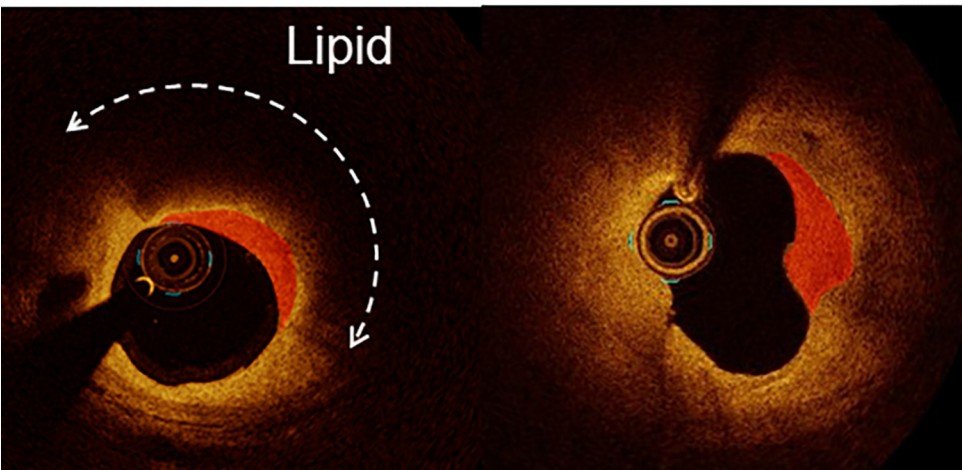

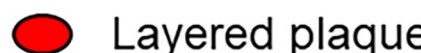

**Fig 2. Optical coherence tomography findings.** A lipidic plaque had a region with strong signal attenuation and diffuse border. Layered plaque was defined as a layer of tissue located close to the luminal surface with clear demarcation from the underlying lipidic plaque (A) or non-lipidic plaque (B).

coronary flow in the mid-distal portion of the LAD was visualized in a modified 3-chamber view. For color-flow mapping, the velocity range was set to 16–24 cm/s. A sample volume (3–5 mm wide) was pointed at the distal LAD to measure blood flow velocity [20]. All data were digitally stored for offline review and measurement. Three optimal flow signal profiles at rest and during hyperemia were obtained offline from the recorded data. The peak diastolic coronary flow velocity was measured at resting conditions (rDPV) and during maximal hyperemia (hyperemic diastolic peak flow velocity [hDPV]), which was induced by intravenous adenosine administration. Coronary flow velocity reserve (CFVR) was calculated as the ratio of hDPV to rDPV [21] Coronary flow improvement was evaluated using the metric %hDPV increase, defined as (post-PCI hDPV minus pre-PCI hDPV)/pre-PCI hDPV × 100. Clinical

demographics and angiographic, physiological, and OCT findings were compared according to the increase or decrease in %hDPV. We additionally explored predictors of %hDPV decrease (defined as <0% %hDPV increase) and higher-level %hDPV increase (defined as >50% %hDPV increase, the highest tertile in this population). The area under the curve (AUC) in receiver-operating characteristics curve (ROC) for hDPV in predicting hyperemic coronary flow decrease was 0.773 (95% confidence interval [CI], 0.664–0.882) (S1 Fig).

### Statistical analysis

Patient clinical demographics are presented as n (%) when appropriate. Categorical data are expressed as absolute frequencies and percentages and were compared using the $\chi^2$ test. Continuous variables are expressed as medians (25th–75th percentile) and compared using the Mann–Whitney U test or Kruskal–Wallis test, as appropriate. Normality of the data was tested using the Shapiro–Wilk test, and all variables were non-normally distributed.

Univariable and multivariable logistic regression analyses were performed to identify the independent preprocedural predictors of a coronary flow decrease or higher-level increase after PCI. Covariates with P <0.10 in the univariable analyses were included in the multivariable analysis, using Akaike's information criterion to avoid overfitting. ROC curve analyses were performed to explore the best cut-off value for each physiological parameter to predict hDPV decrease (S1 Fig) or higher-level hDPV increase (>50%) (S2 Fig).

Intra- and inter-observer variability in the diagnosis of layered plaques by OCT was tested by two independent observers and repeated by an observer 4 weeks apart in 40 randomly selected cases using Kappa statistics. Statistical significance was set at P <0.05. Statistical analyses were performed using R software version 4.3.1 (R Foundation for Statistical Computing, Vienna, Austria).

## Results

### Clinical demographics, angiographic data, and physiological parameters

FFR increased in all cases after PCI (0.68±0.08 to 0.83±0.05 [P<0.01]). Both hDPV and CFVR increased after PCI (hDPV: 55.6±21.7 cm/s to 69.5±20.5 cm/s [P<0.01]); CFVR: (2.05±0.64 to 2.55±0.73 [P<0.01]). The median hDPV increase was 27.3% (6.32%, 59.1%). Although hDPV significantly increased after PCI in the cohort, 19.4% (20/103) patients exhibited a decrease in hDPV (Fig 3). Clinical demographics and angiographic findings showed no significant differences between the two groups divided by the increase or decrease in coronary flow after PCI (Table 1). Notably, post-PCI creatine kinase myocardial band and high-sense troponin I showed no significant differences between the two groups (Table 1).

The pre-PCI guidewire-derived FFR was similar between patients with and without an hDPV decrease (Table 2). Pre-PCI S-TDE-derived physiological data showed that hyperemic DPV and CFVR were higher in patients with decreased %hDPV (Table 2). The hyperemic DPV increase rate showed a significant inverse relationship with pre-PCI FFR (R = - 0.31, P<0.01) and pre-PCI hDPV (R = -0.60, P<0.01) (Fig 4). Although pre-PCI FFR was not predictive of decreased versus increased hDPV (AUC 0.559 [95% CI 0.421–0.697]; S1 Fig), it was predictive of a higher-level hDPV increase (>50%) (AUC 0.634 [95% CI 0.515–0.753]; S2 Fig).

### OCT findings according to the hyperemic coronary flow change

There was good concordance in intra- and inter-observer agreements for the diagnosis of a layered plaque (κ = 0.85 and 0.75, respectively). Pre-PCI OCT findings according to the decrease or increase in %hDPV are shown in Table 3. On OCT, layered plaques were

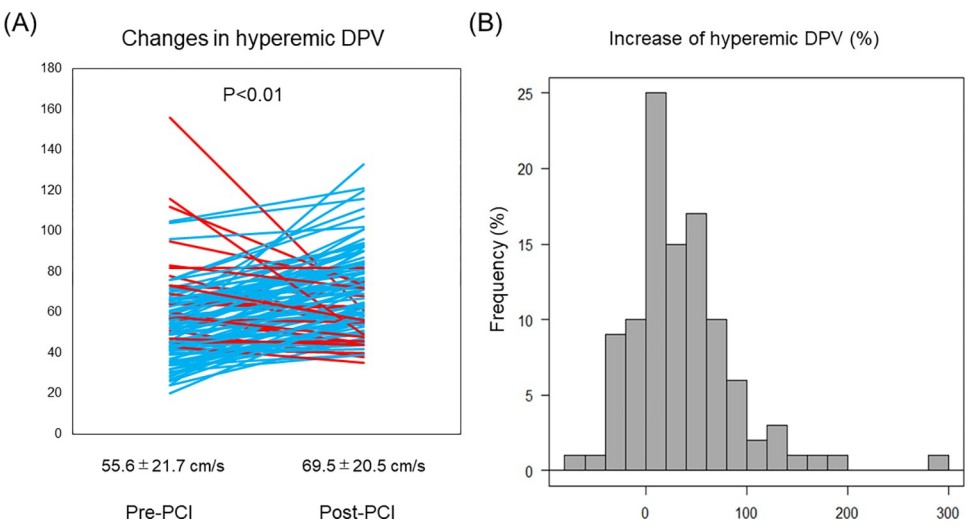

**Fig 3. Changes in hyperemic DPV after PCI.** (A) Overall hyperemic DPV increased after PCI for LAD. (B) Distribution of % hyperemic DPV increase after PCI. Overall, 19% (20/103) patients exhibited % hyperemic DPV decrease. DPV, diastolic peak velocity; PCI, percutaneous coronary intervention.

**Table 1. Clinical demographics and angiographic findings.**

|  | Total (n = 103) | Hyperemic DPV decrease (n = 20) | Hyperemic DPV increase (n = 83) | P-value |
|---|---|---|---|---|
| Clinical demographics | | | | |
| Age, years | 72.0 [60.0, 76.5] | 70.5 [61.8, 75.0] | 72.0 [60.0, 77.0] | 0.66 |
| Male gender | 87 (84.5) | 17 (85.0) | 70 (84.3) | 1.00 |
| Prior myocardial infarction | 32 (31.1) | 5 (25.0) | 27 (32.5) | 0.70 |
| Hypertension | 80 (77.7) | 13 (65.0) | 67 (80.7) | 0.22 |
| Dyslipidemia | 63 (61.2) | 10 (50.0) | 53 (63.9) | 0.38 |
| Diabetes Mellitus | 41 (39.8) | 5 (25.0) | 36 (43.4) | 0.21 |
| Current smoking | 27 (26.2) | 4 (20.0) | 23 (27.7) | 0.67 |
| C-reactive protein, mg/dL | 0.06 [0.03, 0.17] | 0.05 [0.03, 0.19] | 0.06 [0.03, 0.17] | 0.53 |
| Estimated glomerular filtration rate, ml/min/1.73m$^2$ | 66.8 [54.1, 76.8] | 64.9 [54.2, 70.5] | 67.6 [54.1, 77.6] | 0.28 |
| Total cholesterol, mg/dL | 152.0 [125.5, 176.0] | 153.0 [138.0, 166.8] | 152.0 [122.5, 177.0] | 0.76 |
| LDL cholesterol, mg/dL | 75.0 [56.0, 95.0] | 77.5 [61.8, 89.3] | 75.0 [56.0, 100.0] | 0.92 |
| HDL cholesterol, mg/dL | 54.0 [44.3, 65.0] | 59.0 [46.8, 67.5] | 53.5 [43.3, 64.0] | 0.22 |
| Triglycerides, mg/dL | 103.0 [73.0, 148.0] | 98.0 [76.8, 139.0] | 105.0 [71.5, 155.0] | 0.99 |
| Post-PCI creatine kinase myocardial band, IU/L | 7.0 [4.0, 12.5] | 7.0 [4.0, 11.5] | 6.0 [4.0, 12.5] | 0.89 |
| Post-PCI high-sense troponin I, ng/L | 404.0 [173.0, 1284.5] | 401.0 [218.5, 644.8] | 404.0 [147.5, 1339.0] | 0.92 |
| Left ventricular ejection fraction, % | 65.0 [59.5, 70.0] | 61.5 [56.0, 67.5] | 66.0 [60.0, 70.0] | 0.19 |
| Angiographic findings | | | | |
| Minimal lumen diameter, mm | 0.90 [0.73, 1.12] | 0.88 [0.72, 1.04] | 0.90 [0.74, 1.12] | 0.90 |
| Reference diameter, mm | 2.52 [2.17, 3.03] | 2.32 [2.03, 2.58] | 2.65 [2.20, 3.08] | 0.07 |
| Diameter stenosis, % | 64.0 [55.8, 70.7] | 59.7 [55.4, 66.1] | 64.5 [55.8, 71.0] | 0.15 |
| Lesion length, mm | 16.3 [11.9, 25.0] | 16.4 [12.4, 20.5] | 15.7 [11.6, 25.7] | 0.94 |

Values are reported as n (%) or the median (25-75th percentile). DPV, diastolic peak velocity; HDL, high-density lipoprotein; LDL, low-density lipoprotein; PCI, percutaneous coronary intervention.

**Table 2. Physiological parameters.**

|  | Total (n = 103) | Hyperemic DPV decrease (n = 20) | Hyperemic DPV increase (n = 83) | P-value |
|---|---|---|---|---|
| Pre PCI FFR | 0.71 [0.65, 0.74] | 0.71 [0.66, 0.74] | 0.71 [0.63, 0.74] | 0.42 |
| Pre-PCI resting DPV, cm/s | 27.0 [21.0, 32.0] | 31.0 [21.0, 43.3] | 26.0 [20.5, 30.0] | 0.21 |
| Pre-PCI hyperemic DPV, cm/s | 52.0 [42.0, 66.0] | 67.0 [57.0, 79.3] | 49.0 [39.5, 63.0] | <0.01 |
| Pre-PCI CFVR | 2.00 [1.60, 2.37] | 2.50 [1.92, 2.77] | 1.95 [1.55, 2.17] | 0.01 |

Values are reported as n (%) or the median (25-75th percentile). CFVR, coronary flow velocity reserve; DPV, diastolic peak velocity; FFR, fractional flow reserve; PCI, percutaneous coronary intervention.

significantly more frequent in patients with a %hDPV decrease than in those with a %hDPV increase. Other OCT findings showed no significant differences between the groups. The more layered plaques were present in the vessel, the more frequent %hDPV decrease was observed (Fig 5).

### Clinical, angiographic, physiological, and OCT findings according to the presence or absence of layered plaque in the vessel

Clinical demographics and angiographic findings showed no significant differences between patients with and without layered plaques, except for higher triglyceride levels in patients with layered plaques (S1 Table). Although the pre-PCI FFR and S-TDE-derived flow parameters were similar, %hDPV increase tended to be lower in patients with layered plaques (Fig 6 and S2 Table). According to OCT findings, patients with layered plaques were more likely to have lipid-rich plaques (S3 Table).

### Predictors of S-TDE-derived hyperemic coronary flow changes

Multivariable logistic regression analyses showed that pre-PCI S-TDE-derived metrics and OCT-derived layered plaques in the vessels were independently associated with a decrease in hyperemic coronary flow after PCI (Table 4). Regarding the prediction of a higher-level hDPV

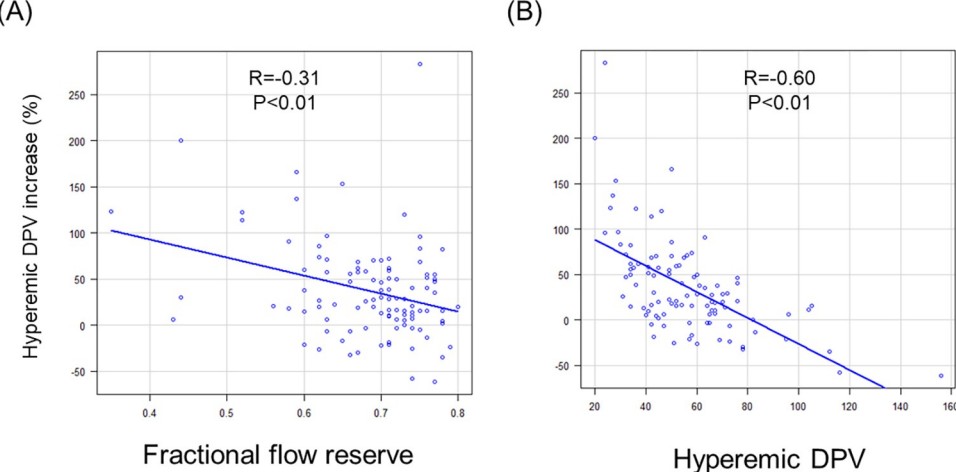

**Fig 4. Relationship between preprocedural physiological parameters and % hyperemic DPV increase.** Percent hyperemic DPV increase was inversely associated with preprocedural fractional flow reserve (A) and hyperemic DPV (B). DPV, diastolic peak velocity.

**Table 3. Optical coherence tomography findings.**

| | Total (n = 103) | Hyperemic DPV decrease (n = 20) | Hyperemic DPV increase (n = 83) | P-value |
|---|---|---|---|---|
| Minimal lumen area, mm$^2$ | 1.34 [1.05, 1.87] | 1.46 [1.00, 1.86] | 1.34 [1.06, 1.90] | 0.99 |
| Thin-cap fibroatheroma | 14 (13.6) | 3 (15.0) | 11 (13.3) | 1.00 |
| Ruptured plaque | 20 (19.4) | 2 (10.0) | 18 (21.7) | 0.38 |
| Lipid-rich plaque | 70 (68.0) | 16 (80.0) | 54 (65.1) | 0.31 |
| Maximum lipid angle, degree | 120.6 [0.0, 183.2] | 139.4 [92.1, 227.1] | 109.7 [0.0, 165.9] | 0.07 |
| Fibrous cap thickness, μm | 120 [90, 225] | 120 [100, 220] | 120 [90, 230] | 0.89 |
| Lipid length, mm | 5.4 [0.0, 9.6] | 6.8 [4.1, 10.7] | 5.30 [0.00, 9.62] | 0.37 |
| Microvessel | 33 (32.0) | 5 (25.0) | 28 (33.7) | 0.62 |
| Macrophage | 85 (82.5) | 20 (100.0) | 65 (78.3) | 0.049 |
| Cholesterol crystal | 41 (39.8) | 9 (45.0) | 32 (38.6) | 0.78 |
| Layered plaque | 59 (57.3) | 17 (85.0) | 42 (50.6) | 0.01 |
| Number of layers in the vessel | | | | <0.01 |
| 0 | 43 (41.7) | 2 (10.0) | 41 (49.4) | |
| 1 | 43 (41.7) | 12 (60.0) | 31 (37.3) | |
| 2 | 15 (14.6) | 4 (20.0) | 11 (13.3) | |
| 3 | 2 (1.9) | 2 (10.0) | 0 (0.0) | |
| Layer on lipidic plaque | 39 (37.9) | 14 (70.0) | 25 (30.1) | <0.01 |
| Layer on non-lipidic plaque | 20 (19.4) | 3 (15.0) | 17 (20.5) | 0.81 |
| Calcification | 75 (72.8) | 14 (70.0) | 61 (73.5) | 0.97 |

Values are reported as n (%) or the median (25-75th percentile). DPV, diastolic peak velocity.

increase (>50%), the presence of layered plaques was not predictive of a higher-level hDPV increase, whereas pre-PCI FFR was predictive of a higher-level hDPV increase independent of pre-PCI rDPV but not of pre-PCI hDPV (Table 5). Our results suggested that the presence of layered plaque and a pre-PCI hDPV ≥57 cm/s predicted coronary flow decrease with a probability of 51.9% even after uncomplicated and successful PCI (Fig 7).

## Discussion

The main findings of this study are as follows. In patients with CCS undergoing successful elective PCI for *de novo* single LAD lesions, (1) overall hDPV significantly increased after PCI, while 19% (20/103) patients exhibited %hDPV decrease; (2) the degree of hyperemic DPV increase showed a significant inverse relation with pre-PCI FFR and pre-PCI hDPV; (3) layered plaques were more frequent in the hDPV-decrease group than in the hDPV-increase group, and this relationship was independent of post-PCI troponin increase; (4) lower pre-PCI FFR value, lower pre-PCI hDPV, and the absence of layered plaque in the vessel were independently associated with the degree of higher-level hDPV increase as continuous variables; and (5) presence of layered plaque and S-TDE-derived metrics were independently predictive of % hDPV decrease after PCI.

In the present study, the pre-PCI FFR was inversely related to hDPV increase, suggesting that compared to lesions with a high FFR, those with a low FFR may benefit more from PCI with respect to an increase in hyperemic coronary flow. These results are consistent with those of previous studies that used a combined pressure and flow velocity wire [9] and phase-contrast cine-magnetic resonance-derived volumetric coronary sinus flow [5] Johnson et al. performed a meta-analysis exploring clinical outcomes after FFR measurement and reported that

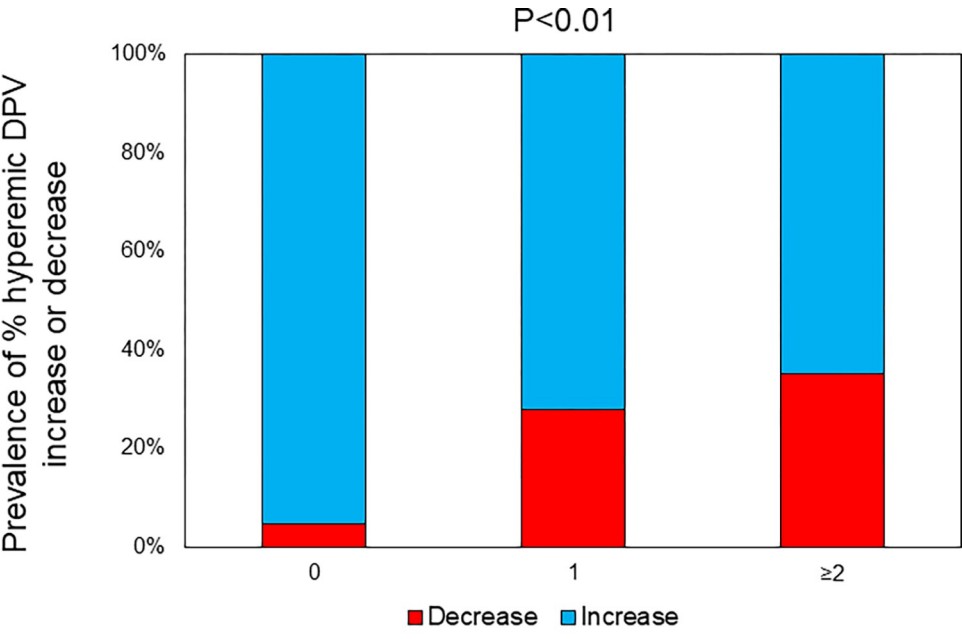

**Fig 5. Association between the number of layered plaques and % hyperemic DPV increase.** The greater the number of layered plaques in the vessel, the more frequent % hyperemic DPV decrease was observed. DPV, diastolic peak velocity.

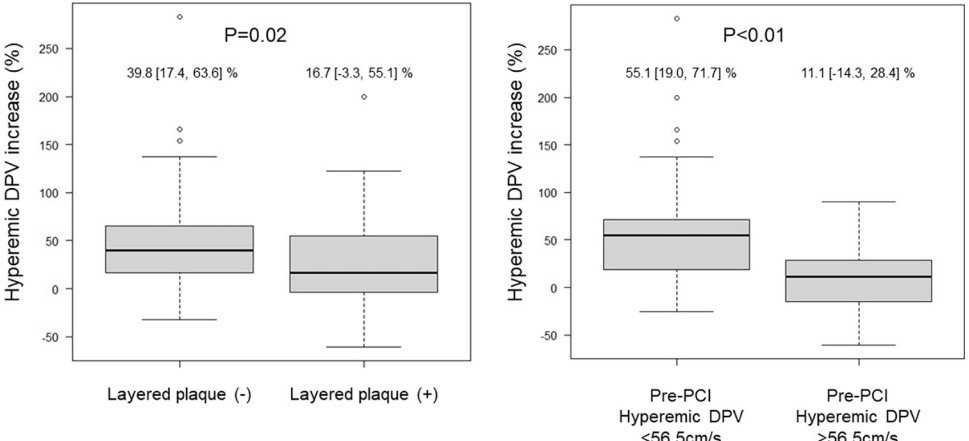

**Fig 6. Percent hyperemic DPV increase according to the preprocedural optical coherence tomography and S-TDE findings.** Percent hyperemic DPV increase was significantly lower in vessels with layered plaques (A) and in vessels with high preprocedural hyperemic DPV (B). DPV, diastolic peak velocity; PCI, percutaneous coronary intervention; S-TDE, stress-transthoracic Doppler echocardiography.

**Table 4. Predictors of coronary hyperemic DPV decrease after percutaneous coronary intervention.**

| | Univariable analysis | | | Multivariable model 1 | | | Multivariable model 2 | | |
|---|---|---|---|---|---|---|---|---|---|
| | OR | 95% CI | P-value | OR | 95% CI | P-value | OR | 95% CI | P-value |
| Male | 1.05 | 0.27–4.11 | 0.94 | - | - | - | - | - | - |
| Hypertension | 0.44 | 0.15–1.29 | 0.14 | - | - | - | - | - | - |
| Left ventricular ejection fraction, % | 0.98 | 0.94–1.02 | 0.36 | - | - | - | - | - | - |
| Pre-PCI FFR, per 0.01 | 1.04 | 0.97–1.12 | 0.25 | - | - | - | - | - | - |
| Pre-PCI resting DPV, cm/s | 1.05 | 1.01–1.11 | 0.03 | 1.05 | 1.00–1.11 | 0.04 | - | - | - |
| Pre-PCI hyperemic DPV, cm/s | 1.05 | 1.02–1.08 | <0.01 | - | - | - | 1.06 | 1.02–1.09 | <0.01 |
| Presence of layered plaque in the vessel | 5.53 | 1.51–20.3 | 0.01 | 5.57 | 1.49–20.9 | 0.01 | 6.94 | 1.61–30.0 | <0.01 |

CI, confidence interval; DPV, diastolic peak velocity; FFR, fractional flow reserve; OR, odds ratio; PCI, percutaneous coronary intervention.

the net benefit of revascularization was achieved with lower baseline FFR values, showing 0.67 as an FFR threshold to guide the absolute benefit of revascularization [22] The threshold was similar to the best cut-off value of pre-PCI FFR to predict a higher-level %hDPV increase (>50%) after PCI in the present study (S2 Fig).

The presence of layered plaques was associated with a decrease in hyperemic coronary flow after PCI. Pathologically, an OCT-derived layered plaque represents an organizing thrombus in its healing stage following episodic previous thrombotic events due to subclinical plaque rupture or erosion in the vessel [17, 23] Recently, we reported an association between OCT-derived layered plaques and unrecognized myocardial infarction detected by late-gadolinium enhancement in patients with CCS and no history of known myocardial infarction [24] suggesting the clinical significance of a layered plaque as a signature of the downstream infarcted myocardium. Given that the microvasculature in an infarct territory is structurally or functionally impaired due to distal embolization, release of inflammatory and vasoconstrictor substances, and formation of platelet-neutrophil aggregates [25] layered plaques might be concomitant with the deteriorated autoregulative function of the microvasculature downstream. Additionally, our results showing that the impact of layered plaques on coronary flow decrease was mainly driven by layered plaques on lipids may suggest an additional influence of adjacent lipid-rich components on downstream coronary flow impairment after elective PCI.

## Clinical implications

In the present study, coronary flow decrease was predicted by the presence of layered plaques and a high pre-PCI hyperemic DPV with a modest positive predictive value (52%) and a high

**Table 5. Predictors of higher-level coronary hyperemic DPV increase after PCI (>50% increase).**

| | Univariable analysis | | | Multivariable model 1 | | | Multivariable model 2 | | |
|---|---|---|---|---|---|---|---|---|---|
| | OR | 95% CI | P-value | OR | 95% CI | P-value | OR | 95% CI | P-value |
| Male | 4.07 | 0.87–19.1 | 0.07 | 3.22 | 0.63–16.6 | 0.16 | 4.39 | 0.69–27.8 | 0.12 |
| Hypertension | 2.85 | 0.89–9.18 | 0.08 | 4.74 | 1.26–17.8 | 0.02 | 4.25 | 0.93–19.4 | 0.06 |
| Left ventricular ejection fraction, % | 1.01 | 0.97–1.05 | 0.65 | - | - | - | - | - | - |
| Pre-PCI FFR, per 0.01 | 0.94 | 0.89–0.99 | 0.02 | 0.94 | 0.89–0.99 | 0.04 | 0.96 | 0.89–1.03 | 0.22 |
| Pre-PCI resting DPV, cm/s | 0.93 | 0.89–0.99 | 0.01 | 0.93 | 0.87–0.98 | 0.01 | - | - | - |
| Pre-PCI hyperemic DPV, cm/s | 0.91 | 0.87–0.95 | <0.01 | - | - | - | 0.91 | 0.87–0.95 | <0.01 |
| Presence of layered plaque in the vessel | 0.54 | 0.23–1.23 | 0.14 | - | - | - | - | - | - |

CI, confidence interval; DPV, diastolic peak velocity; FFR, fractional flow reserve; OR, odds ratio; PCI, percutaneous coronary intervention.

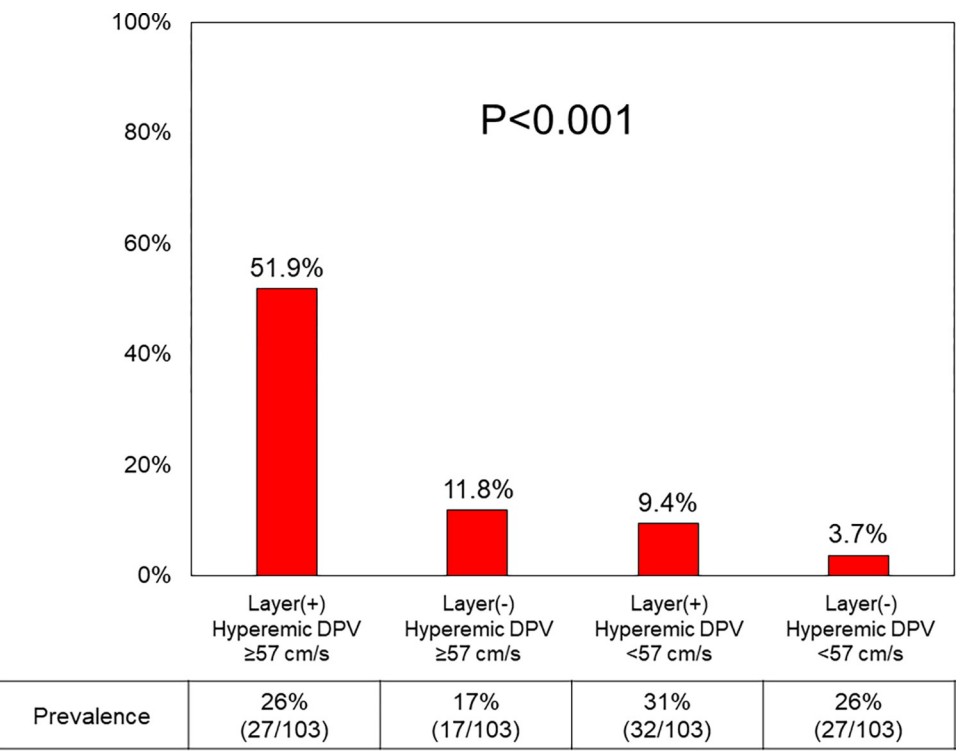

| Prevalence | 26%<br>(27/103) | 17%<br>(17/103) | 31%<br>(32/103) | 26%<br>(27/103) |
| --- | --- | --- | --- | --- |

**Fig 7. Prevalence of hyperemic coronary flow decrease.** The prevalence of coronary flow decrease in the four groups was categorized by the presence of layered plaque and the best cutoff value of preprocedural hyperemic DPV. DPV, diastolic peak velocity.

negative predictive value (96%). Thus, OCT imaging of flow-limiting vessels showing no layered plaques could be a potential marker of beneficial PCI with respect to the expected goal of increasing coronary flow. Our approach using multimodal physiological and imaging tools, including OCT, FFR, and noninvasive S-TDE, can potentially identify coronary lesions that may or may not benefit from revascularization by optimizing the coronary flow increase. The present study supports the potential efficacy of identifying layered plaques in predicting a decrease in coronary flow after elective PCI, enabling further optimization of contemporary FFR-guided PCI. Additionally, noninvasive predictors of the presence of layered plaques may help identify coronary lesions that benefit from elective PCI before catheterization. As our results are merely hypothesis-generating, further prospective exploration with clinical outcome data is needed.

## Study limitations

This study has some limitations. First, it included patients from a single center and had a retrospective design, both of which make selection bias unavoidable. Second, the decision to perform a physiological assessment was made at the discretion of the operator, which is another potential source of selection bias. Third, we analyzed S-TDE only in the LAD artery, limiting the implications of the study. Fourth, the present study included no prognostic data in terms of the degree of %hDPV increase, suggesting its limited contribution to clinical outcomes. Fifth, we performed S-TDE color-flow mapping only in distal LAD by apical modified 3-chamber view. Further attempts of flow visualization including parasternal scanning may provide further noninvasive information of coronary flow impairment and stenosis assessment of various LAD segments [26].

## Conclusions

In patients undergoing successful uncomplicated elective PCI for *de novo* single LAD lesions, layered plaques detected by OCT were significantly more frequent in the hDPV-decrease group than in the hDPV-increase group, independent of post-PCI cardiac marker elevation. Therefore, plaque morphology and S-TDE-derived metrics may help identify lesions which benefit from elective PCI.

## Supporting information

**S1 Data.**
(XLSX)

**S1 Fig. Prediction of hyperemic coronary flow decrease.** Receiver-operating characteristic curve analysis to predict %hDPV decrease. AUC, area under the curve; CI, confidence interval; DPV, diastolic peak velocity; FFR, fractional flow reserve; PCI, percutaneous coronary intervention.
(TIF)

**S2 Fig. Prediction of higher-level hyperemic coronary flow increase.** Receiver-operating characteristic curve analysis to predict higher-level %hDPV increase ($>50\%$). AUC, area under the curve; CI, confidence interval; DPV, diastolic peak velocity; FFR, fractional flow reserve; PCI, percutaneous coronary intervention.
(TIF)

**S1 Table. Clinical demographics and angiographic findings according to the presence or absence of layered plaque in the vessel.** Values are reported as n (%) or the median (25-75th percentile). DPV, diastolic peak velocity; HDL, high-density lipoprotein; LDL, low-density lipoprotein; PCI, percutaneous coronary intervention.
(DOCX)

**S2 Table. Physiological parameters according to the presence or absence of layered plaque in the vessel.** Values are reported as n (%) or the median (interquartile range). CFVR, coronary flow velocity reserve; DPV, diastolic peak velocity; FFR, fractional flow reserve; PCI, percutaneous coronary intervention; $T_{mn}$, mean transit time.
(DOCX)

**S3 Table. Optical coherence tomographic findings according to the presence or absence of layered plaque in the vessel.** Values are reported as n (%) or the median (25-75th percentile).
(DOCX)

## Acknowledgments

The authors thank all the clinical technicians in the Department of Clinical Laboratory at Tsuchiura Kyodo General Hospital for their assistance in obtaining the echocardiographic data of the patients involved in the study.

## Author Contributions

**Conceptualization:** Eisuke Usui, Tsunekazu Kakuta.

**Data curation:** Eisuke Usui.

**Formal analysis:** Eisuke Usui.

**Investigation:** Eisuke Usui, Yoshihiro Hanyu, Tatsuya Sakamoto, Masahiro Hoshino, Masahiro Hada, Tatsuhiro Nagamine, Kai Nogami, Hiroki Ueno, Mirei Setoguchi, Kazuki Matsuda, Kodai Sayama, Tomohiro Tahara, Takashi Mineo, Yoshihisa Kanaji, Tomoyo Sugiyama, Taishi Yonetsu.

**Methodology:** Eisuke Usui.

**Project administration:** Eisuke Usui.

**Resources:** Eisuke Usui, Yoshihiro Hanyu, Tatsuya Sakamoto, Masahiro Hoshino, Masahiro Hada, Tatsuhiro Nagamine, Kai Nogami, Hiroki Ueno, Mirei Setoguchi, Kazuki Matsuda, Kodai Sayama, Tomohiro Tahara, Takashi Mineo, Yoshihisa Kanaji, Tomoyo Sugiyama, Taishi Yonetsu.

**Supervision:** Taishi Yonetsu, Tetsuo Sasano, Tsunekazu Kakuta.

**Visualization:** Eisuke Usui.

**Writing – original draft:** Eisuke Usui.

**Writing – review & editing:** Tsunekazu Kakuta.

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
