## [Decision Letter · Decision Letter 0]

7 May 2024

PONE-D-24-11473Association between optical coherence tomography-defined culprit morphologies and changes in hyperemic coronary flow after elective stenting assessed by transthoracic Doppler echocardiographyPLOS ONE

Dear Dr. Usui,

Thank you for submitting your manuscript to PLOS ONE. After careful consideration, we feel that it has merit but does not fully meet PLOS ONE’s publication criteria as it currently stands. Therefore, we invite you to submit a revised version of the manuscript that addresses the points raised during the review process.

Please address each item from both reviewers in separate comments in the response letter. 

We look forward to receiving your revised manuscript.

Kind regards,

R. Jay Widmer

Academic Editor

PLOS ONE

Journal Requirements:

"NO authors have competing interests"

Additional Editor Comments:

The reviewers were generally favorable regarding this manuscript. There are some methodological and data reporting items to address that should help to strengthen this manuscript.

Reviewers' comments:

Reviewer's Responses to Questions

**Comments to the Author**

1. Is the manuscript technically sound, and do the data support the conclusions?

Reviewer #1: Yes

Reviewer #2: No

2. Has the statistical analysis been performed appropriately and rigorously? 

Reviewer #1: Yes

Reviewer #2: No

3. Have the authors made all data underlying the findings in their manuscript fully available?

Reviewer #1: Yes

Reviewer #2: Yes

4. Is the manuscript presented in an intelligible fashion and written in standard English?

Reviewer #1: Yes

Reviewer #2: Yes

5. Review Comments to the Author

Reviewer #1: Dear editor,

first of all thank you for asking me to review this paper. This is an interesting and well written manuscript, whose aim is to identify the association between optical coherence tomography (OCT) defined culprit morphology and changes in hyperemic coronary flow, comparing values before and after angioplasty on left anterior descending (LAD).

Besides fractional flow reserve (FFR) during the procedure, a stress transthoracic doppler echocardiography (S-TDE) was performed before and after angioplasty, with particular attention to peak diastolic coronary flow velocity at rest and after maximal hyperemia. The results are appealing and confirm that the presence of a layered plaque was independently associated with hyperemic coronary flow decrease as assessed by S-TDE. The clinical implications are astonishing considering that a multimodal approach can potentially identify coronary lesions that may or not benefit from revascularization.

For that regards the content:

- Is sample size sufficient? If yes, please explain how it has been calculated.

- Which segment of LAD (proximal, mid, distal) was considered? Was S-TDE equally performant for all segments of LAD?

In literature other authors have stressed that transthoracic enhanced color Doppler (E-Doppler TTE) could help to identify relevant stenosis: for instance Caiati et al (doi 10.3389/fcvm.2023.1186983) have recently reported that accelerated stenotic flow in the entire LAD, assessed by transthoracic enhanced color Doppler can reveal coronary stenosis and its severity. Please evaluate this article for potential citation

Reviewer #2: The study by Usui et al. evaluates the association between optical coherence tomography-defined culprit morphologies and changes in hyperemic coronary flow assessed by transthoracic Doppler echocardiography after elective PCI. Introduction and design of the study are well written, however, the results are confused and scattered.

To improve the quality of the study, I suggest the following:

1. The section “Physiological assessment”, should be explained with more details.

2. In the section „S-TDE acquisition and analysis”, sentences regarding rAVP and hAVP are unnecessary. Therefore, the third diagram regarding hAPV in both Figure S1 and Figure 2 is also unnecessary, as well as the appropriate rows in Tables 2, 4 and 5. In the Table 2, row “Guidewire-derived parameters” and row “Stress-transthoracic Doppler echocardiographic parameters” should be also deleted.

3. I do not understand the term “higher-level %hDPV increase”?

4. In the whole manuscript, as well as in all tables, angiographical term “Minimum lumen diameter” should be replaced with the term “Minimal luminal diameter”.

5. Figure S3 is unnecessary.

6. Table 4 is incorrect and should be replaced with Table S4 (without APV measurements).

6. PLOS authors have the option to publish the peer review history of their article (what does this mean?). If published, this will include your full peer review and any attached files.

Reviewer #1: No

Reviewer #2: No

---

## [Author Response · Author response to Decision Letter 0]

27 May 2024

We thank the reviewers for insightful and constructive comments. We have tried to address all the issues and suggestions raised by the reviewers, and a point-by-point response letter has been provided below. We believe that the manuscript has improved and hope that the revised version meets the requirements by the editor and the reviewers.

Reviewer #1: Dear editor,

first of all thank you for asking me to review this paper. This is an interesting and well written manuscript, whose aim is to identify the association between optical coherence tomography (OCT) defined culprit morphology and changes in hyperemic coronary flow, comparing values before and after angioplasty on left anterior descending (LAD).

Besides fractional flow reserve (FFR) during the procedure, a stress transthoracic doppler echocardiography (S-TDE) was performed before and after angioplasty, with particular attention to peak diastolic coronary flow velocity at rest and after maximal hyperemia. The results are appealing and confirm that the presence of a layered plaque was independently associated with hyperemic coronary flow decrease as assessed by S-TDE. The clinical implications are astonishing considering that a multimodal approach can potentially identify coronary lesions that may or not benefit from revascularization.

Response: We appreciate the reviewer’s time and effort in reviewing our manuscript. We tried to address all the issues raised by the reviewer and provided our point-by-point responses below.

For that regards the content:

- Is sample size sufficient? If yes, please explain how it has been calculated.

Response: We thank the reviewer for raising an important issue. Because this study is of retrospective and observational nature, initially no sample size calculation needed to draw statistically powered results was conducted. Given that there have been no established data of cutoff values, prevalences and sampling errors regarding significant transthoracic echocardiography-derived hyperemic coronary flow change after PCI, it was considered challenging to estimate appropriate sample size. As described in the section of limitation, our results are merely hypothesis-generating, and further studies are required to validate our results and its clinical significance. 

- Which segment of LAD (proximal, mid, distal) was considered? Was S-TDE equally performant for all segments of LAD?

In literature other authors have stressed that transthoracic enhanced color Doppler (E-Doppler TTE) could help to identify relevant stenosis: for instance Caiati et al (doi 10.3389/fcvm.2023.1186983) have recently reported that accelerated stenotic flow in the entire LAD, assessed by transthoracic enhanced color Doppler can reveal coronary stenosis and its severity. Please evaluate this article for potential citation

Response: We thank the reviewer for pointing out an important issue and suggesting an article which introduces new methods to obtain coronary flow in various segments of LAD. In this study, we performed apical modified 3-chamber view to obtain LAD flow, not attempting parasternal scanning to obtain left main, LAD proximal or mid flow as Caiati et al. provided in the article suggested by the reviewer. Thus, we assumed that all LAD flows in our study were tranced in distal segments of LAD, as previously reported in the ASE guideline (Pellikka PA et al. J Am Soc Echocardiogr 2020;33:1-41). According to the reviewer’s suggestion, we added the article in Study limitation section and added two references as shown below.

Page 23, Line 341; “Fifth, we performed S-TDE color-flow mapping only in distal LAD by apical modified 3-chamber view. Further attempts of flow visualization including parasternal scanning may provide further noninvasive information of coronary flow impairment and stenosis assessment of various LAD segments [26].”

Page 28, Line 435; “20. Pellikka PA, Arruda-Olson A, Chaudhry FA, Chen MH, Marshall JE, Porter TR, et al. Guidelines for Performance, Interpretation, and Application of Stress Echocardiography in Ischemic Heart Disease: From the American Society of Echocardiography. Journal of the American Society of Echocardiography. 2020;33: 1-41.e8. doi:10.1016/j.echo.2019.07.001”

Page 29, Line 458; “26. Caiati C, Pollice P, Iacovelli F, Sturdà F, Lepera ME. Accelerated stenotic flow in the left anterior descending coronary artery explains the causes of impaired coronary flow reserve: an integrated transthoracic enhanced Doppler study. Front Cardiovasc Med. 2023;10. doi:10.3389/fcvm.2023.1186983”

Reviewer: 2

The study by Usui et al. evaluates the association between optical coherence tomography-defined culprit morphologies and changes in hyperemic coronary flow assessed by transthoracic Doppler echocardiography after elective PCI. Introduction and design of the study are well written, however, the results are confused and scattered.

Response: We appreciate the reviewer’s time and effort in reviewing our manuscript. We tried to do our best to address all the issues raised by the reviewer and provided the point-by-point responses below. 

To improve the quality of the study, I suggest the following:　

1. The section “Physiological assessment”, should be explained with more details.

Response: We thank the reviewer for pointing out the issue. Since we only measured FFR by guidewire in this population, the core part of the section was simple. We revised the description to be more detailed according to the suggestion by the reviewer.

Page 5, Line 121; “Hyperemia was induced by an intravenous infusion of adenosine 5’-triphosphate (160 μg/kg/min). The FFR was calculated by dividing the mean distal pressure (Pd) by the mean aortic pressure (Pa) during steady-state maximal hyperemia. After FFR assessment, when the pressure sensor reached the tip of the guiding catheter during hyperemia via a pull-back maneuver, a mean Pd–Pa pressure drift of ≤ 2 mmHg was documented. We mandated repeat assessment if the pressure drift was > 2 mmHg. All patients were instructed to strictly refrain from caffeinated beverages for > 24 hours before catheterization.”

2. In the section “S-TDE acquisition and analysis”, sentences regarding rAVP and hAVP are unnecessary. Therefore, the third diagram regarding hAPV in both Figure S1 and Figure 2 is also unnecessary, as well as the appropriate rows in Tables 2, 4 and 5. In the Table 2, row “Guidewire-derived parameters” and row “Stress-transthoracic Doppler echocardiographic parameters” should be also deleted.

Response: According to the reviewer’s suggestion, we deleted results of APVs from Tables, Figures and the manuscript.

3. I do not understand the term “higher-level %hDPV increase”?

Response: We agree that the definition of “higher-level %hDPV increase” was ambiguous in the original manuscript. The threshold was >50% which was the highest tertile of %hDPV increase in this population. We added the explanation in the revised manuscript as shown below.

Page 8, Line 178; “We additionally explored predictors of %hDPV decrease (defined as <0% %hDPV increase) and higher-level %hDPV increase (defined as >50% %hDPV increase, the highest tertile of DPV changes in this population).”

4. In the whole manuscript, as well as in all tables, angiographical term “Minimum lumen diameter” should be replaced with the term “Minimal luminal diameter”.

Response: According to the reviewer’s suggestion, we replaced “Minimum lumen diameter” with “Minimal lumen diameter”.

5. Figure S3 is unnecessary.

Response: As the reviewer commented, we deleted figure S3.

6. Table 4 is incorrect and should be replaced with Table S4 (without APV measurements).

Response: According to the reviewer’s suggestion, we replaced Table 4 with Table S4 (revised Table 5).

---

## [Decision Letter · Decision Letter 1]

4 Jul 2024

Association between optical coherence tomography-defined culprit morphologies and changes in hyperemic coronary flow after elective stenting assessed by transthoracic Doppler echocardiography

PONE-D-24-11473R1

Dear Dr. Kakuta,

We’re pleased to inform you that your manuscript has been judged scientifically suitable for publication and will be formally accepted for publication once it meets all outstanding technical requirements.

Kind regards,

R. Jay Widmer

Academic Editor

PLOS ONE

Additional Editor Comments (optional):

The authors have addressed many of the concerns from the reviewers.

Reviewers' comments:

Reviewer's Responses to Questions

**Comments to the Author**

1. If the authors have adequately addressed your comments raised in a previous round of review and you feel that this manuscript is now acceptable for publication, you may indicate that here to bypass the “Comments to the Author” section, enter your conflict of interest statement in the “Confidential to Editor” section, and submit your "Accept" recommendation.

Reviewer #1: All comments have been addressed

Reviewer #2: All comments have been addressed

2. Is the manuscript technically sound, and do the data support the conclusions?

Reviewer #1: Yes

Reviewer #2: No

3. Has the statistical analysis been performed appropriately and rigorously? 

Reviewer #1: Yes

Reviewer #2: No

4. Have the authors made all data underlying the findings in their manuscript fully available?

Reviewer #1: Yes

Reviewer #2: Yes

5. Is the manuscript presented in an intelligible fashion and written in standard English?

Reviewer #1: Yes

Reviewer #2: No

6. Review Comments to the Author

Reviewer #1: (No Response)

Reviewer #2: The manuscript contains fundamental errors that cannot be rectified through author revisions. The presenting results, all tables are unclear and confusing, and should be significantly improved.

7. PLOS authors have the option to publish the peer review history of their article (what does this mean?). If published, this will include your full peer review and any attached files.

Reviewer #1: No

Reviewer #2: No

---

## [Editor Report · Acceptance letter]

5 Aug 2024

PONE-D-24-11473R1 

PLOS ONE

Dear Dr. Kakuta, 

I'm pleased to inform you that your manuscript has been deemed suitable for publication in PLOS ONE. Congratulations! Your manuscript is now being handed over to our production team.

Kind regards, 

on behalf of

Dr. R. Jay Widmer 

Academic Editor

PLOS ONE